# Dieback of *Euonymus alatus* (Celastraceae) Caused by *Cytospora haidianensis* sp. nov. in China

**Xian Zhou, Meng Pan, Haoyu Li, Chengming Tian and Xinlei Fan \***

The Key Laboratory for Silviculture and Conservation of Ministry of Education, Beijing Forestry University, Beijing 100083, China; xianz1001@163.com (X.Z.); 15652651577@163.com (M.P.); zjn0702@gmail.com (H.L.); chengmt@bjfu.edu.cn (C.T.)

**\*** Correspondence: xinleifan@bjfu.edu.cn; Tel.: +86-130-2120-4929

**Abstract:** *Euonymus alatus* (Celastraceae) is widely cultivated in China for its economic value and landscape benefits. *Euonymus alatus* dieback occurs due to members of *Cytospora* and has become one of the most severe diseases affecting its cultivation in China. In this study, we examined the causal agent of bough dieback on campuses of University Road, Beijing, China. Among the strains, three were morphologically consistent with *Cytospora*, showing hyaline and allantoid conidia. Based on phylogenetic analyses of the concatenated actin (ACT), internal transcribed spacer (ITS), RNA polymerase II second largest subunit (RPB2), translation elongation factor 1-alpha (TEF1-$\alpha$) and beta-tubulin (TUB2) gene sequences, along with morphological and physiological features, we propose *C. haidianensis* as a novel species. It was confirmed as a causal agent of dieback of *E. alatus* by pathogenicity tests. Mycelial growth of *Cytospora haidianensis* occurred at pH values ranging from 3.0 to 11.0, with optimum growth at 8.3, and at temperatures from 5 to 35 °C, with optimum growth at 19.8 °C. We also tested the growth of *C. haidianensis* in the presence of six carbon sources. Sucrose, maltose and glucose were highly efficient and xylose was the least. The ability of *C. haidianensis* to grow at 19.8 °C may help to explain its occurrence causing dieback of *E. alatus* in Beijing during the autumn season.

**Keywords:** Cytosporaceae; Diaporthales; mycelial growth; pathogenicity; phylogeny; taxonomy

## 1. Introduction

*Euonymus alatus* (Celastraceae) has been widely cultivated for ornamental landscaping in China because of its tolerance to many environmental conditions [1]. At present, the related research on the fungal diseases of *Euonymus* is mainly on anthracnose caused by *Colletotrichum gloeosporioides*, powdery mildew by *Oidium euonymi-japonici* and dieback by *Cytospora euonymicola* and *C. euonymina* [2,3].

The genus *Cytospora* has wide distribution and has often been regarded as comprising phytopathogens, endophytes or saprobes occurring on a broad range of hosts [3,4]. Several species have been reported as pathogens causing severe branch or trunk dieback disease on monocotyledonous, dicotyledonous and gymnosperm hosts (e.g., Anacardiaceae, Elaeagnaceae, Fabaceae, Juglandaceae, Myrtaceae, Rosaceae, Salicaceae and Ulmaceae) [5,6]. The symptoms of *Cytospora* canker are elongate, slightly sunken and discoloured areas in the bark at first, then the forming of several prominent black fruit bodies [5]. Conidia emerge from the fructifications in the form of yellow to orange or red gelatinous tendrils under moist conditions [3]. *Cytospora* species have single or multiple locules (and/or diaporthalean-like perithecia), filamentous conidiophores (and/or clavate to elongate obovoid asci) and allantoid hyaline conidia (and/or ascospores) [5]. As plant pathogens, *Cytospora* species have also been reported to be associated with other diseases, such as root rot of Chinese jujube and collar rot of pomegranate [7,8].

In the past, it was difficult to name *Cytospora* species because of their morphological overlap, causing confusion in species delimitation. Previously, identification of *Cytospora* species was mainly based on host affiliations, often with unclear morphological descriptions. Since the advent of molecular analysis, morphology and phylogeny using internal transcribed spacer (ITS) sequence data were combined to describe 28 species of *Cytospora* from *Eucalyptus*, of which 11 species were new to science [5]. Later, similar methods were used to describe 14 species from South Africa [6]. However, only ITS sequences are available for most known *Cytospora* species, ex-type sequence data are available for only a few species and many taxa need epitypification. Thus, recent studies have subsequently emphasized only part of *Cytospora* species using a polyphasic approach to solve the confusion in species recognition [3,4,7,8].

Stem and branch dieback have occurred on *Euonymus alatus* growing on the streets of campuses of University Road in Beijing, China. Typical symptoms of the disease are stem blight and dieback, with lesions extending along the entire branch. Infected stems have light brown to brown pigmentation. According to our observation, the disease seriously affects the colour of *Euonymus* plants and growth status, along with significant damage to the landscape. The aim of this study was to identify the causal agent causing *E. alatus* dieback disease based on molecular, morphological and physiological data.

## 2. Materials and Methods

### 2.1. Collection and Isolation

Three diseased branches of *E. alatus* were selected from 20 infected plants observed during collecting trips on the campuses of University Road in Beijing, China. Part of the hymenium containing 3 to 4 fruiting bodies of fresh material was cut horizontally with a sterile blade and crushed in a drop of sterile water on a glass slide. The contents were agitated with the blade until a spore suspension was obtained. Half of the spore suspension was then spread over the surface of 1.8% potato dextrose agar (PDA) in a petri dish and incubated at 25 °C for up to 24 h, and a single germinating conidium was transferred to a fresh PDA plate. Specimens were deposited at the working collection of X.L. Fan (CF) housed at Beijing Forestry University (BJFC) and living cultures were deposited at the China Forestry Culture Collection Centre (CFCC).

### 2.2. Morphological Observation

Specimens were observed on infected plant tissues, and the structure and size of fruiting bodies, the presence or absence of a conceptacle, and the size and shape of the spores were recorded. Macro-morphological photographs were captured using a Leica stereomicroscope (M205), including size of conidiomata; the presence or absence of special structures such as conceptacle and central column; number and diameter of ostioles per ectostromatic disc; colour, shape and size of discs; and number of locules. Micro-morphological observations such as size and shape of conidiophores and conidia were determined under a Nikon Eclipse 80i microscope equipped with a Nikon digital sight DS-Ri2 high-definition colour camera with differential interference contrast (DIC). Over 30 conidiomata were sectioned and 50 conidia were selected randomly to measure their lengths and widths. Colony diameters were measured, and the colony colours were described after 3 and 14 days according to the colour charts of Rayner [9]. Adobe Bridge CS6 and Adobe Photoshop CS6 were used for manual editing.

### 2.3. DNA Extraction, PCR Amplification and Sequencing

Mycelium for DNA extraction was grown on PDA with cellophane for 3 days and obtained from the surface of the cellophane by scraping. Genomic DNA was extracted using the modified CTAB method [10]. DNA concentrations were estimated visually by electrophoresis in 1% agarose gel by comparing band intensity with a DNA marker 1 kbp (Takara Bio USA, Inc., Mountain View, CA, USA). PCR amplifications were performed in a DNA Engine (PTC-200) Peltier Thermal Cycler

(Bio-Rad Laboratories, Hercules, CA, USA). DNA was amplified from actin (ACT), internal transcribed spacer (ITS), RNA polymerase II second largest subunit (RPB2), translation elongation factor 1-alpha (TEF1-$\alpha$) and beta-tubulin (TUB2) following Fan et al. [3]. The ACT region was amplified using primers ACT-512F and ACT-1567R [11]. The ITS rDNA region was amplified and sequenced with primers ITS-1 and ITS-4 [12]. The RPB2 was amplified with primers RPB2-5F and fRPB2-7cR [13]. The TEF1-$\alpha$ was amplified with primers EF1-688F and EF1-986R [11,14]. The TUB2 was amplified with primers Bt-2a and Bt-2b [15]. The PCR amplification products were electrophoresed and visualized in gels. The DNA sequencing was performed using an ABI PRISM® 3730XL DNA Analyzer with BigDye® Terminator Kit v.3.1 (Invitrogen) at the Shanghai Invitrogen Biological Technology Company Limited (Beijing, China). DNA sequences generated by each primer combination were used to obtain consensus sequences using Seqman v.7.1 and the DNASTAR Lasergene Core Suite software package (DNASTAR Inc., Madison, WI, USA).

### 2.4. Phylogenetic Analysis

The current isolates were initially identified as *Cytospora* sp. based on morphological observations and BLAST results. To clarify their further phylogenetic position, an analysis based on the 5 combined genes (ACT, ITS, RPB2, TEF1-$\alpha$ and TUB2) was constructed to compare *Cytospora* species from the current study with other strains in the GenBank database. *Diaporthe vaccinii* CBS 160.32 was selected as the outgroup in all analyses. Subsequent alignments for each gene were generated using MAFFT v.7 [16] and manually adjusted using MEGA v.6 [17]. Ambiguously aligned sequences were excluded from the analysis. Reference sequences were selected based on ex-type or ex-epitype sequences available from recently published literature [5,7,18–24] (Table 1).

Phylogenetic analyses were formed by PAUP v.4.0b10 for the maximum parsimony (MP) method [25], MrBayes v.3.1.2 for the Bayesian inference (BI) method [26] and RAxML v.7.2.8 for the maximum likelihood (ML) method [27]. Tree length (TL), consistency index (CI), retention index (RI) and rescaled consistency (RC) were calculated [25]. ML analysis was generated using a GTR+G+I model of site substitution following recent study [4], including estimation of gamma distributed rate heterogeneity and proportion of invariant sites [27]. Branch support was evaluated with a bootstrapping method of 1000 replicates [28]. BI analysis was performed using a Markov chain Monte Carlo (MCMC) algorithm with Bayesian posterior probabilities [29]. A nucleotide substitution model was estimated by MrModeltest v.2.3 [30] and a weighted Bayesian analysis was considered. Two MCMC chains were run from random trees for 10,000,000 generations and trees were sampled each 100th generation. The first 40% of trees were discarded as the burn-in phase of each analysis and the Bayesian posterior probability (BPP) was calculated to assess the remaining trees [29]. The branch support from MP and ML analysis was evaluated with a bootstrapping (BS) method of 1000 replicates [28]. Phylograms were constructed using Figtree v.1.3.1 [31]. Sequence data were deposited in GenBank. The aligned matrices used for phylogenetic analysis were submitted through TreeBASE (www.treebase.org; study ID S26000).

### 2.5. Pathogenicity Test

Three *Cytospora* strains (CFCC 54184, CFCC 54056 and CFCC 54057) obtained in this study were used to conduct the pathogenicity test. The pathogenicity test was performed on 1-year-old *E. alatus* plants obtained from seeds kept in a greenhouse at constant 28 °C and 99% relative humidity. On healthy plants, twigs to be used for inoculation were surface disinfected with 75% ethanol for 1 min. The bark surface of each disinfected twig was scalded with a sterilized inoculating loop within a region 5 mm in length to a depth of 2 mm. For mycelial inoculation, a 5 mm diameter PDA plug with mycelium was taken from a 3-day-old colony and inoculated onto the wounded twigs. Three replicates were conducted for each isolate. Non-colonized PDA plugs and sterile water were used as negative controls. Pathogenicity was determined by the length of the necrotic lesion caused by the tested isolates, which was measured 3 weeks after inoculation. Fungal isolates were re-isolated from the infected tissue, and morphological characterization and DNA sequence comparisons were conducted to follow Koch's postulates.

**Table 1.** Strains and GenBank accession numbers of *Cytospora* species used in the phylogenetic analyses in this study.

| Species | Strain[1] | Host | Origin | GenBank Accession Numbers | | | | |
|---|---|---|---|---|---|---|---|---|
| | | | | ACT | ITS | RPB2 | TEF1-$\alpha$ | TUB2 |
| *Cytospora ailanthicola* | CFCC 89970[T] | *Ailanthus altissima* | China | MH933526 | MH933618 | MH933592 | MH933494 | MH933565 |
| *Cytospora leucosperma* | CFCC 89622 | *Pyrus bretschneideri* | China | KU710988 | KR045616 | KU710944 | KU710911 | KR045657 |
| | CFCC 89894 | *Pyrus bretschneideri* | China | KU710989 | KR045617 | KU710945 | KU710912 | KR045658 |
| *Cytospora ampulliformis* | MFLUCC 16-0583[T] | *Sorbus intermedia* | Russia | KY417692 | KY417726 | KY417794 | NA | NA |
| | MFLUCC 16-0629 | *Acer platanoides* | Russia | KY417693 | KY417727 | KY417795 | NA | NA |
| *Cytospora amygdali* | CBS 144233[T] | *Prunus dulcis* | USA | MG972002 | MG971853 | NA | MG971659 | MG971718 |
| *Cytospora atrocirrhata* | CFCC 89615 | *Juglans regia* | China | KF498673 | KR045618 | KU710946 | KP310858 | KR045659 |
| | CFCC 89616 | *Juglans regia* | China | KF498674 | KR045619 | KU710947 | KP310859 | KR045660 |
| *Cytospora beilinensis* | CFCC 50493[T] | *Pinus armandii* | China | MH933527 | MH933619 | NA | MH933495 | MH933561 |
| | CFCC 50494 | *Pinus armandii* | China | MH933528 | MH933620 | NA | MH933496 | MH933562 |
| *Cytospora berberidis* | CFCC 89927[T] | *Berberis dasystachya* | China | KU710990 | KR045620 | KU710948 | KU710913 | KR045661 |
| | CFCC 89933 | *Berberis dasystachya* | China | KU710991 | KR045621 | KU710949 | KU710914 | KR045662 |
| *Cytospora bungeana* | CFCC 50495[T] | *Pinus bungeana* | China | MH933529 | MH933621 | MH933593 | MH933497 | MH933563 |
| | CFCC 50496 | *Pinus bungeana* | China | MH933530 | MH933622 | MH933594 | MH933498 | MH933564 |
| *Cytospora californica* | CBS 144234[T] | *Juglans regia* | USA | MG972083 | MG971935 | NA | MG971645 | NA |
| *Cytospora carbonacea* | CFCC 89947 | *Ulmus pumila* | China | KP310842 | KR045622 | KU710950 | KP310855 | KP310825 |
| *Cytospora carpobroti* | CMW 48981[T] | *Carpobrotus edulis* | South Africa | NA | MH382812 | NA | MH411212 | MH411207 |
| *Cytospora castanae* | DBT 183[T] | *Castanea sativa* | North India | NA | KC963921 | NA | NA | NA |
| *Cytospora celtidicola* | CFCC 50497[T] | *Celtis sinensis* | China | MH933531 | MH933623 | MH933595 | MH933499 | MH933566 |
| | CFCC 50498 | *Celtis sinensis* | China | MH933532 | MH933624 | MH933596 | MH933500 | MH933567 |
| *Cytospora centrivillosa* | MFLUCC 16-1206[T] | *Sorbus domestica* | Italy | NA | MF190122 | MF377600 | NA | NA |
| | MFLUCC 17-1660 | *Sorbus domestica* | Italy | NA | MF190123 | MF377601 | NA | NA |

**Table 1.** *Cont.*

| Species | Strain[1] | Host | Origin | GenBank Accession Numbers | | | | |
|---|---|---|---|---|---|---|---|---|
| | | | | ACT | ITS | RPB2 | TEF1-$\alpha$ | TUB2 |
| *Cytospora ceratosperma* | CFCC 89624 | *Juglans regia* | China | NA | KR045645 | KU710976 | KP310860 | KR045686 |
| | CFCC 89625 | *Juglans regia* | China | NA | KR045646 | KU710977 | KP31086 | KR045687 |
| *Cytospora ceratospermopsis* | CFCC 89626[T] | *Juglans regia* | China | KU711011 | KR045647 | KU710978 | KU710934 | KR045688 |
| | CFCC 89627 | *Juglans regia* | China | KU711012 | KR045648 | KU710979 | KU710935 | KR045689 |
| *Cytospora chrysosperma* | CFCC 89629 | *Salix psammophila* | China | NA | KF765673 | KF765705 | NA | NA |
| | CFCC 89981 | *Populus alba* subsp. *pyramidalis* | China | MH933533 | MH933625 | MH933597 | MH933501 | MH933568 |
| | CFCC 89982 | *Ulmus pumila* | China | KP310835 | KP281261 | NA | KP310848 | KP310818 |
| *Cytospora coryli* | CFCC 53162T | *Corylus mandshurica* | China | NA | MN854450 | MN850751 | MN850758 | MN861120 |
| *Cytospora cotini* | MFLUCC 14-1050[T] | *Cotinus coggygria* | Russia | NA | KX430142 | KX430144 | NA | NA |
| *Cytospora curvata* | MFLUCC 15-0865[T] | *Salix alba* | Russia | KY417694 | KY417728 | KY417796 | NA | NA |
| *Cytospora davidiana* | CXY 1350[T] | *Populus davidiana* | China | NA | KM034870 | NA | NA | NA |
| | CXY 1374 | *Populus davidiana* | China | NA | KM034869 | NA | NA | NA |
| *Cytospora diopuiensis* | MFLUCC 18-1419[T] | Undefined wood | Thailand | MN685819 | MK912137 | NA | NA | NA |
| *Cytospora leucostoma* | MFLUCC 15-0864 | *Crataegus monogyna* | Ukraine | KY417729 | KY417729 | KY41769 | KY417797 | NA |
| *Cytospora elaeagni* | CFCC 89632 | *Elaeagnus angustifolia* | China | KU710995 | KR045626 | KU710955 | KU710918 | KR045667 |
| | CFCC 89633 | *Elaeagnus angustifolia* | China | KU710996 | KF765677 | KU710956 | KU710919 | KR045668 |
| *Cytospora elaeagnicola* | CFCC 52882[T] | *Elaeagnus angustifolia* | China | MK732344 | MK732341 | MK732347 | NA | NA |
| | CFCC 52883 | *Elaeagnus angustifolia* | China | MK732345 | MK732342 | MK732348 | NA | NA |
| | CFCC 52884 | *Elaeagnus angustifolia* | China | MK732346 | MK732343 | MK732349 | NA | NA |
| *Cytospora erumpens* | CFCC 50022 | *Prunus padus* | China | MH933534 | MH933627 | NA | MH933502 | MH933569 |
| | MFLUCC 16-0580[T] | *Salix × fragilis* | Russia | KY417699 | KY417733 | KY417801 | NA | NA |
| *Cytospora eucalypti* | CBS 144241 | *Eucalyptus globulus* | USA | MG972056 | MG971907 | NA | MG971617 | MG971772 |

**Table 1.** *Cont.*

| Species | Strain[1] | Host | Origin | GenBank Accession Numbers | | | | |
|---|---|---|---|---|---|---|---|---|
| | | | | ACT | ITS | RPB2 | TEF1-$\alpha$ | TUB2 |
| *Cytospora euonymicola* | CFCC 50499[T] | *Euonymus kiautschovicus* | China | MH933535 | MH933628 | MH933598 | MH933503 | MH933570 |
| | CFCC 50500 | *Euonymus kiautschovicus* | China | MH933536 | MH933629 | MH933599 | MH933504 | MH933571 |
| *Cytospora euonymina* | CFCC 89993[T] | *Euonymus kiautschovicus* | China | MH933537 | MH933630 | MH933600 | MH933505 | MH933590 |
| | CFCC 89999 | *Euonymus kiautschovicus* | China | MH933538 | MH933631 | MH933601 | MH933506 | MH933591 |
| *Cytospora fraxinigena* | MFLUCC 14-0868[T] | *Fraxinus ornus* | Italy | NA | MF190133 | NA | NA | NA |
| | MFLU 17-0880 | *Fraxinus ornus* | Italy | NA | MF190134 | NA | NA | NA |
| *Cytospora fugax* | CXY 1371 | *Populus simonii* | China | NA | KM034852 | NA | NA | KM034891 |
| | CXY 1381 | *Populus ussuriensis* | China | NA | KM034853 | NA | NA | KM034890 |
| *Cytospora galegicola* | MFLUCC 18-1199[T] | *Galega officinalis* | Italy | MN685810 | MK912128 | MN685820 | NA | NA |
| *Cytospora germanica* | CXY 1322 | *Elaeagnus oxycarpa* | China | NA | JQ086563 | NA | NA | NA |
| *Cytospora gigalocus* | CFCC 89620[T] | *Juglans regia* | China | KU710997 | KR045628 | KU710957 | KU710920 | KR045669 |
| | CFCC 89621 | *Juglans regia* | China | KU710998 | KR045629 | KU710958 | KU710921 | KR045670 |
| *Cytospora gigaspora* | CFCC 50014 | *Juniperus procumbens* | China | KU710999. | KR045630 | KU710959 | KU710922 | KR045671 |
| | CFCC 89634[T] | *Salix psammophila* | China | KU711000 | KF765671 | KU710960 | KU710923 | KR045672 |
| *Cytospora granati* | CBS 144237[T] | *Punica granatum* | USA | MG971949 | MG971799 | NA | MG971514 | MG971664 |
| ***Cytospora haidianensis*** | **CFCC 54056** | ***Euonymus alatus*** | **China** | **MT363978** | **MT360041** | **MT363987** | **MT363997** | **MT364007** |
| | **CFCC 54057[T]** | ***Euonymus alatus*** | **China** | **MT363979** | **MT360042** | **MT363988** | **MT363998** | **MT364008** |
| | **CFCC 54184** | ***Euonymus alatus*** | **China** | **MT363980** | **MT360043** | **MT363989** | **MT363999** | **MT364009** |
| *Cytospora hippophaës* | CFCC 89639 | *Hippophaë rhamnoides* | China | KU711001 | KR045632 | KU710961 | KU710924 | KR045673 |
| | CFCC 89640 | *Hippophaë rhamnoides* | China | KF765730 | KF765682 | KU710962 | KP310865 | KR045674 |
| *Cytospora japonica* | CFCC 89956 | *Prunus cerasifera* | China | KU710993 | KR045624 | KU710953 | KU710916 | KR045665 |
| | CFCC 89960 | *Prunus cerasifera* | China | KU710994 | KR045625 | KU710954 | KU710917 | KR045666 |
| *Cytospora joaquinensis* | CBS 144235[T] | *Populus deltoides* | USA | MG972044 | MG971895 | NA | MG971605 | MG971761 |

**Table 1.** *Cont.*

| Species | Strain[1] | Host | Origin | GenBank Accession Numbers | | | | |
|---|---|---|---|---|---|---|---|---|
| | | | | ACT | ITS | RPB2 | TEF1-$\alpha$ | TUB2 |
| *Cytospora junipericola* | BBH 42444 | *Juniperus communis* | Italy | NA | MF190126 | NA | MF377579 | NA |
| | MFLU 17-0882[T] | *Juniperus communis* | Italy | NA | MF190125 | NA | MF377580 | NA |
| *Cytospora juniperina* | CFCC 50501[T] | *Juniperus przewalskii* | China | MH933539 | MH933632 | MH933602 | MH933507 | NA |
| | CFCC 50502 | *Juniperus przewalskii* | China | MH933540 | MH933633 | MH933603 | MH933508 | MH933572 |
| | CFCC 50503 | *Juniperus przewalskii* | China | MH933541 | MH933634 | MH933604 | MH933509 | NA |
| *Cytospora kantschavelii* | CXY 1383 | *Populus maximowiczii* | China | NA | KM034867 | NA | NA | NA |
| | CXY 1386 | *Populus maximowiczii* | China | NA | KM034867 | NA | NA | NA |
| *Cytospora kuanchengensis* | CFCC 52464[T] | *Castanea mollissima* | China | MK442940 | MK432616 | MK578076 | NA | NA |
| | CFCC 52465 | *Castanea mollissima* | China | MK442941 | MK432617 | MK578077 | NA | NA |
| *Cytospora leucostoma* | CFCC 50015 | *Sorbus aucuparia* | China | KU711002 | KR045634 | NA | KU710925 | KR045675 |
| | CFCC 50016 | *Sorbus aucuparia* | China | MH820408 | MH820400 | NA | MH820404 | MH820389 |
| | CFCC 50017 | *Prunus cerasifera* | China | MH933542 | MH933635 | NA | MH933510 | MH933573 |
| | CFCC 50018 | *Prunus serrulata* | China | MH933543 | MH933636 | NA | MH933511 | MH933574 |
| | CFCC 50019 | *Rosa helenae* | China | MH933544 | MH933637 | NA | NA | NA |
| | CFCC 50020 | *Prunus persica* | China | MH933545 | MH933638 | NA | NA | NA |
| | CFCC 50021 | *Prunus salicina* | China | MH933546 | MH933639 | NA | MH933512 | MH933575 |
| | CFCC 50023 | *Cornus alba* | China | KU711003 | KR045635 | KU710964 | KU710926 | KR045676 |
| | CFCC 50024 | *Prunus pseudocerasus* | China | MH933547 | MH933640 | MH933605 | NA | MH933576 |
| | CFCC 50467 | *Betula platyphylla* | China | NA | KT732948 | NA | NA | NA |
| | CFCC 50468 | *Betula platyphylla* | China | NA | KT732949 | NA | NA | NA |
| | CFCC 53140 | *Prunus sibirica* | China | MN850760 | MN854445 | MN850746 | MN850753 | MN861115 |
| | CFCC 53141 | *Prunus sibirica* | China | MN850761 | MN854446 | MN850747 | MN850754 | MN861116 |
| | CFCC 53156 | *Juglans mandshurica* | China | MN850762 | MN854447 | MN850748 | MN850755 | MN861117 |
| | MFLUCC 16-0574 | *Rosa* sp. | Russia | KY417696 | KY417731 | KY417798 | NA | NA |

**Table 1.** *Cont.*

| Species | Strain[1] | Host | Origin | GenBank Accession Numbers | | | | |
|---------|-----------|------|--------|------|------|------|------|------|
| | | | | ACT | ITS | RPB2 | TEF1-α | TUB2 |
| *Cytospora longiostiolata* | MFLUCC 16-0628[T] | *Salix × fragilis* | Russia | KY417700 | KY417734 | KY417802 | NA | NA |
| *Cytospora longispora* | CBS 144236[T] | *Prunus domestica* | USA | MG972054 | MG971905 | NA | MG971615 | MG971764 |
| *Cytospora lumnitzericola* | MFLUCC 17-0508[T] | *Lumnitzera racernosa* | Thailand | MH253457 | MG975778 | MH253453 | NA | NA |
| *Cytospora mali* | CFCC 50028 | *Malus pumila* | China | MH933548 | MH933641 | MH933606 | MH933513 | MH933577 |
| | CFCC 50029 | *Malus pumila* | China | MH933549 | MH933642 | MH933607 | MH933514 | MH933578 |
| | CFCC 50030 | *Malus pumila* | China | MH933550 | MH933643 | MH933608 | MH933524 | MH933579 |
| | CFCC 50031 | *Crataegus* sp. | China | KU711004 | KR045636 | KU710965 | KU710927 | KR045677 |
| | CFCC 50044 | *Malus baccata* | China | KU711005 | KR045637 | KU710966 | KU710928 | KR045678 |
| *Cytospora melnikii* | CFCC 89984 | *Rhus typhina* | China | MH933551 | MH933644 | MH933609 | MH933515 | MH933580 |
| | MFLUCC 15-0851[T] | *Malus domestica* | Russia | KY417701 | KY417735 | KY417803 | NA | NA |
| | MFLUCC 16-0635 | *Populus nigra* var. *italica* | Russia | KY417702 | KY417736 | KY417804 | NA | NA |
| *Cytospora myrtagena* | CFCC 52454 | *Castanea mollissima* | China | MK442938 | MK432614 | MK578074 | NA | NA |
| | CFCC 52455 | *Castanea mollissima* | China | MK442939 | MK432615 | MK578075 | NA | NA |
| *Cytospora nivea* | MFLUCC 15-0860 | *Salix acutifolia* | Russia | KY417703 | KY417737 | KY417805 | NA | NA |
| | CFCC 89641 | *Elaeagnus angustifolia* | China | KU711006 | KF765683 | KU710967 | KU710929 | KR045679 |
| | CFCC 89643 | *Salix psammophila* | China | NA | KF765685 | KU710968 | KP310863 | KP310829 |
| *Cytospora notastroma* | NE_TFR5 | *Populus tremuloides* | USA | NA | JX438632 | NA | JX438543 | NA |
| | NE_TFR8 | *Populus tremuloides* | USA | NA | JX438633 | NA | JX438542 | NA |
| *Cytospora oleicola* | CBS 144248[T] | *Olea europaea* | USA | MG972098 | MG971944 | NA | MG971660 | MG971752 |
| *Cytospora palm* | CXY 1276 | *Cotinus coggygria* | China | NA | JN402990 | NA | KJ781296 | NA |
| | CXY 1280[T] | *Cotinus coggygria* | China | NA | JN411939 | NA | KJ781297 | NA |
| *Cytospora parakantschavelii* | MFLUCC 15-0857[T] | *Populus × sibirica* | Russia | KY417704 | KY417738 | KY417806 | NA | NA |
| | MFLUCC 16-0575 | *Pyrus pyraster* | Russia | KY417705 | KY417739 | KY417807 | NA | NA |

**Table 1.** *Cont.*

| Species | Strain[1] | Host | Origin | GenBank Accession Numbers | | | | |
|---------|-----------|------|--------|------|------|------|--------|------|
| | | | | **ACT** | **ITS** | **RPB2** | **TEF1-α** | **TUB2** |
| *Cytospora parapistaciae* | CBS 144506[T] | *Pistacia vera* | USA | MG971954 | MG971804 | NA | MG971519 | MG971669 |
| *Cytospora parasitica* | MFLUCC 15-0507[T] | *Malus domestica* | Russia | KY417706 | KY417740 | KY417808 | NA | NA |
| | XJAU 2542-1 | *Malus* sp. | China | NA | MH798884 | NA | MH813452 | NA |
| *Cytospora paratranslucens* | MFLUCC 15-0506[T] | *Populus alba* var. *bolleana* | Russia | KY417707 | KY417741 | KY417809 | NA | NA |
| | MFLUCC 16-0627 | *Populus alba* | Russia | KY417708 | KY417742 | KY417810 | NA | NA |
| *Cytosporapiceae* | CFCC 52841T | *Picea crassifolia* | China | MH820406 | MH820398 | MH820395 | MH820402 | MH820387 |
| | CFCC 52842 | *Picea crassifolia* | China | MH820407 | MH820399 | MH820396 | MH820403 | MH820388 |
| *Cytospora pingbianensis* | MFLUCC 18-1204[T] | Undefined wood | China | MN685817 | MK912135 | MN685826 | NA | NA |
| *Cytospora pistaciae* | CBS 144238[T] | *Pistacia vera* | USA | MG971952 | MG971802 | NA | MG971517 | MG971667 |
| *Cytospora platanicola* | MFLU 17-0327[T] | *Platanus hybrida* | Italy | MH253449 | MH253451 | MH253450 | NA | NA |
| *Cytospora platyclada* | CFCC 50504[T] | *Platycladus orientalis* | China | MH933552 | MH933645 | MH933610 | MH933516 | MH933581 |
| | CFCC 50505 | *Platycladus orientalis* | China | MH933553 | MH933646 | MH933611 | MH933517 | MH933582 |
| | CFCC 50506 | *Platycladus orientalis* | China | MH933554 | MH933647 | MH933612 | MH933518 | MH933583 |
| *Cytospora platycladicola* | CFCC 50038[T] | *Platycladus orientalis* | China | MH933555 | KT222840 | MH933613 | MH933519 | MH933584 |
| | CFCC 50039 | *Platycladus orientalis* | China | KU711008 | KR045642 | KU710973 | KU710931 | KR045683 |
| *Cytospora plurivora* | CBS 144239[T] | *Olea europaea* | USA | MG972010 | MG971861 | NA | MG971572 | MG971726 |
| *Cytospora populicola* | CBS 144240[T] | *Populus deltoides* | USA | MG972040 | MG971891 | NA | MG971601 | MG971757 |
| *Cytospora populina* | CFCC 89644[T] | *Salix psammophila* | China | KU711007 | KF765686 | KU710969 | KU710930 | KR045681 |
| *Cytospora populinopsis* | CFCC 50032[T] | *Sorbus aucuparia* | China | MH933556 | MH933648 | MH933614 | MH933520 | MH933585 |
| | CFCC 50033 | *Sorbus aucuparia* | China | MH933557 | MH933649 | MH933615 | MH933521 | MH933586 |
| *Cytospora pruinopsis* | CFCC 50034[T] | *Ulmus pumila* | China | KP310836 | KP281259 | KU710970 | KP310849 | KP310819 |
| | CFCC 50035 | *Ulmus pumila* | China | KP310837 | KP281260 | KU710971 | KP310850 | KP310820 |
| | CFCC 53153 | *Ulmus pumila* | China | MN850763 | MN854451 | MN850752 | MN850759 | MN861121 |

**Table 1.** *Cont.*

| Species | Strain[1] | Host | Origin | GenBank Accession Numbers | | | | |
|---------|-----------|------|--------|------|------|------|--------|------|
| | | | | ACT | ITS | RPB2 | TEF1-$\alpha$ | TUB2 |
| *Cytospora predappioensis* | MFLUCC 17-2458[T] | *Platanus hybrida* | Italy | NA | MG873484 | NA | NA | NA |
| *Cytospora pruinosa* | CFCC 50036 | *Syringa oblata* | China | KP310832 | KP310800 | NA | KP310845 | KP310815 |
| | CFCC 50037 | *Syringa oblata* | China | MH933558 | MH933650 | NA | MH933522 | MH933589 |
| *Cytospora prunicola* | MFLU 17-0995[T] | *Prunus* sp. | Italy | MG742353 | MG742350 | MG742352 | NA | NA |
| *Cytospora pubescentis* | MFLUCC 18-1201[T] | *Quercus pubescens* | Italy | MN685812 | MK912130 | MN685821 | NA | NA |
| *Cytospora punicae* | CBS 144244 | *Punica granatum* | USA | MG972091 | MG971943 | NA | MG971654 | MG971798 |
| *Cytospora quercicola* | MFLUCC 14-0867[T] | *Quercus* sp. | Italy | NA | MF190129 | NA | NA | NA |
| | MFLU 17-0881 | *Quercus* sp. | Italy | NA | MF190128 | NA | NA | NA |
| *Cytospora ribis* | CFCC 50026 | *Ulmus pumila* | China | KP310843 | KP281267 | KU710972 | KP310856 | KP310826 |
| | CFCC 50027 | *Ulmus pumila* | China | KP310844 | KP281268 | NA | KP310857 | KP310827 |
| *Cytospora rosae* | MFLU 17-0885 | *Rosa canina* | Italy | NA | MF190131 | NA | NA | NA |
| *Cytospora rostrata* | CFCC 89909[T] | *Salix cupularis* | China | KU711009 | KR045643 | KU710974 | KU710932 | KR045684 |
| | CFCC 89910 | *Salix cupularis* | China | KU711010 | KR045644 | KU710975 | KU710933 | NA |
| *Cytospora rusanovii* | MFLUCC 15-0853 | *Populus × sibirica* | Russia | KY417709 | KY417743 | KY417811 | NA | NA |
| | MFLUCC 15-0854[T] | *Salix babylonica* | Russia | KY417710 | KY417744 | KY417812 | NA | NA |
| *Cytospora salicacearum* | MFLUCC 15-0861 | *Salix × fragilis* | Russia | KY417711 | KY417745 | KY417813 | NA | NA |
| | MFLUCC 15-0509[T] | *Salix alba* | Russia | KY417712 | KY417746 | KY417814 | NA | NA |
| | MFLUCC 16-0576 | *Populus nigra* var. *italica* | Russia | KY417707 | KY417741 | KY417809 | NA | NA |
| | MFLUCC 16-0587 | *Prunus cerasus* | Russia | KY417708 | KY417742 | KY417810 | NA | NA |
| *Cytospora salicicola* | MFLUCC 15-0866 | *Salix alba* | Russia | KY417715 | KY417749 | KY417817 | NA | NA |
| | MFLUCC 14-1052[T] | *Salix alba* | Russia | KU982637 | KU982636 | NA | NA | NA |
| *Cytospora salicina* | MFLUCC 15-0862[T] | *Salix alba* | Russia | KY417716 | KY417750 | KY417818 | NA | NA |
| | MFLUCC 16-0637 | *Salix × fragilis* | Russia | KY417717 | KY417751 | KY417819 | NA | NA |

**Table 1.** *Cont.*

| Species | Strain[1] | Host | Origin | GenBank Accession Numbers | | | | |
|---------|-----------|------|--------|------|-----|------|--------|------|
| | | | | ACT | ITS | RPB2 | TEF1-$\alpha$ | TUB2 |
| *Cytospora schulzeri* | CFCC 50040 | *Malus domestica* | China | KU711013 | KR045649 | KU710980 | KU710936 | KR045690 |
| | CFCC 50042 | *Malus asiatica* | China | KU711014 | KR045650 | KU710981 | KU710937 | KR045691 |
| *Cytospora sibiraeae* | CFCC 50045[T] | *Sibiraea angustata* | China | KU711015 | KR045651 | KU710982 | KU710938 | KR045692 |
| | CFCC 50046 | *Sibiraea angustata* | China | KU711015 | KR045652 | KU710983 | KU710939 | KR045693 |
| *Cytospora sophorae* | CFCC 50047 | *Styphnolobium japonicum* | China | KU711017 | KR045653 | KU710984 | KU710940 | KR045694 |
| | CFCC 50048 | *Magnolia grandiflora* | China | MH820409 | MH820401 | MH820397 | MH820405 | MH820390 |
| | CFCC 89598 | *Styphnolobium japonicum* | China | KU711018 | KR045654 | KU710985 | KU710941 | KR045695 |
| *Cytospora sophoricola* | CFCC 89595[T] | *Styphnolobium japonicum* var. *pendula* | China | KU711019 | KR045655 | KU710986 | KU710942 | KR045696 |
| | CFCC 89596 | *Styphnolobium japonicum* var. *pendula* | China | KU711020 | KR045656 | KU710987 | KU710943 | KR045697 |
| *Cytospora sophoriopsis* | CFCC 89600[T] | *Styphnolobium japonicum* | China | KU710992 | KR045623 | KU710951 | KU710915 | KP310817 |
| *Cytospora sorbi* | MFLUCC 16-0631[T] | *Sorbus aucuparia* | Russia | KY417718 | KY417752 | KY417820 | NA | NA |
| *Cytospora sorbicola* | MFLUCC 16-0584[T] | *Acer pseudoplatanus* | Russia | KY417721 | KY417755 | KY417823 | NA | NA |
| | MFLUCC 16-0633 | *Cotoneaster melanocarpus* | Russia | KY417724 | KY417758 | KY417826 | NA | NA |
| *Cytospora spiraeae* | CFCC 50049[T] | *Spiraea salicifolia* | China | MG708196 | MG707859 | MG708199 | NA | NA |
| | CFCC 50050 | *Spiraea salicifolia* | China | MG708197 | MG707860 | MG708200 | NA | NA |
| *Cytospora spiraeicola* | CFCC 53138[T] | *Spiraea salicifolia* | China | NA | MN854448 | MN850749 | MN850756 | MN861118 |
| | CFCC 53139 | *Tilia nobilis* | China | NA | MN854449 | NA | NA | NA |
| *Cytospora tamaricicola* | CFCC 50507 | *Rosa multifolora* | China | MH933559 | MH933651 | MH933616 | MH933525 | MH933587 |
| | CFCC 50508[T] | *Tamarix chinensis* | China | MH933560 | MH933652 | MH933617 | MH933523 | MH933588 |
| *Cytospora tanaitica* | MFLUCC 14-1057[T] | *Betula pubescens* | Russia | KT459413 | KT459411 | NA | NA | NA |

**Table 1.** *Cont.*

| Species | Strain[1] | Host | Origin | GenBank Accession Numbers | | | | |
|---|---|---|---|---|---|---|---|---|
| | | | | ACT | ITS | RPB2 | TEF1-$\alpha$ | TUB2 |
| *Cytospora thailandica* | MFLUCC 17-0262[T] | *Xylocarpus moluccensis* | Thailand | MH253459 | MG975776 | MH253455 | NA | NA |
| | MFLUCC 17-0263[T] | *Xylocarpus moluccensis* | Thailand | MH253460 | MG975777 | MH253456 | NA | NA |
| *Cytospora tibouchinae* | CPC 26333[T] | *Tibouchina semidecandra* | France | NA | KX228284 | NA | NA | NA |
| *Cytospora translucens* | CXY 1351 | *Populus davidiana* | China | NA | KM034874 | NA | NA | KM034895 |
| *Cytospora ulmi* | MFLUCC 15-0863[T] | *Ulmus minor* | Russia | NA | KY417759 | NA | NA | NA |
| *Cytospora vinacea* | CBS 141585[T] | *Vitis interspecific* hybrid 'Vidal' | USA | NA | KX256256 | NA | KX256277 | KX256235 |
| *Cytospora xinglongensis* | CFCC 52458 | *Castanea mollissima* | China | MK442946 | MK432622 | MK578082 | NA | NA |
| | CFCC 52459 | *Castanea mollissima* | China | MK442947 | MK432623 | MK578083 | NA | NA |
| *Cytospora viridistroma* | CBS 202.36[T] | *Cercis canadensis* Castigl. | USA | NA | MN172408 | NA | MN271853 | NA |
| *Cytospora viticola* | CBS 141586[T] | *Vitis vinifera* | USA | NA | KX256239 | NA | KX256260 | KX256218 |
| *Cytospora xylocarpi* | MFLUCC 17-0251[T] | *Xylocarpus granatum* | Thailand | MH253458 | MG975775 | MH253454 | NA | NA |
| *Diaporthe vaccinii* | CBS 160.32 | *Vaccinium macrocarpon* | USA | JQ807297 | KC343228 | NA | KC343954 | KC344196 |

BBH, BIOTEC Bangkok Herbarium, National Science and Technology Development Agency, Thailand; CBS, Westerdijk Fungal Biodiversity Institute (CBS-KNAW Fungal Biodiversity Centre), Utrecht, Netherlands; CFCC, China Forestry Culture Collection Centre, Beijing, China; CMW, culture collection of Michael Wingfield, University of Pretoria, South Africa; CPC, culture collection of Pedro Crous, Netherlands; MFLU, Mae Fah Luang University herbarium, Thailand; MFLUCC, Mae Fah Luang University Culture Collection, Thailand; XJAU, Xinjiang Agricultural University, Xinjiang, China; NA, not applicable. All new isolates used in this study are indicated in bold type and strains from generic type species are marked by a superscript T.

*2.6. Temperature and pH Tests*

The 3 *Cytospora* isolates showed similar growth characteristics, so we used the type strain of the new species (CFCC 54057) to evaluate the effects of temperature and pH on colony growth using PDA plates. Tested temperatures ranged from 0 to 40 °C at intervals of 5 °C (i.e., 0, 5, 10, 15, 20, 25, 30, 35 and 40 °C). In order to clarify the effect of pH on radial mycelial growth, PDA medium was adjusted with 0.1 M NaOH and 0.1 M HCl to obtain pH values from 2.0 to 12.0 at intervals of 1.0 (i.e., 2.0, 3.0, 4.0, 5.0, 6.0, 7.0, 8.0, 9.0, 10.0, 11.0 and 12.0). A 5 mm diameter mycelial plug was placed in the centre of a 90 mm petri dish with PDA medium and incubated at 28 °C in the dark, with 3 replicates for each treatment. The effects of pH and temperature on mycelial growth were determined by measuring the colony diameter after 24, 48, 72 and 96 h of incubation and the data were converted to radial growth in millimetres [32]. Data were analysed in IBM SPSS Statistics v.22.0 (IBM Inc., Armonk, NY, USA) to select the model that best fit the individual data points, and SPSS was used to confirm the selected model. The optimal temperature and pH value of the regression curves were calculated based on the regression equations generated by IBM SPSS Statistics, and output figures with Origin v.8.0.

*2.7. Carbon Colony Growth Test*

To investigate the utilization of carbon sources, the type strain of the new species (CFCC 54057) was incubated in the dark at 28 °C on PDA medium for 4 days. PDA medium was used as the base medium (potato 20 g, sucrose 20 g, agar 17 g, distilled water to complete 1000 mL). The 20 g of sucrose was replaced by 20 g of fructose, galactose, glucose, maltose, sucrose or xylose to test these compounds as carbon sources. A 5 mm diameter PDA plug of mycelium was transferred to the centre of each sole carbon source medium. Colony growth was determined by measuring the colony diameters after incubation for 24, 48, 72 and 96 h at 28 °C in the dark, and the results were subsequently converted to radial growth [32]. Mean comparisons were conducted using Tukey's honestly significant difference (HSD) test ($\alpha$ = 0.05) in SigmaPlot v.14.0.

## 3. Results

*3.1. Phylogenetic Analyses*

A combined matrix of five gene sequences of *Cytospora* species was constructed. The combined alignment matrices (ACT, ITS, RPB2, TEF1-$\alpha$ and TUB2) included 192 accessions (3 from this study and 189 retrieved from GenBank) and counted 3056 characters including gaps (350 characters for ACT, 631 for ITS, 726 for RPB2, 725 for TEF1-$\alpha$ and 624 for TUB2), of whih 1594 characters were constant, 130 variable characters were parsimony-uninformative and 1349 (44.14%) characters were variable and parsimony-informative. The MP analysis generated 200 parsimonious trees, the first of which is presented in Figure 1 (TL = 8,573, CI = 0.312, RI = 0.788, RC = 0.246). The tree topologies of ML and BI analyses were similar to that of the MP tree.

Based on the initial analysis, a second, more inclusive combined matrix was constructed using 27 accessions from the first dataset. The second combined alignment matrix counted 2531 characters including gaps (274 characters for ACT, 529 for ITS, 726 for RPB2, 553 for TEF1-$\alpha$ and 449 for TUB2). In total, 1,819 characters were constant, 182 variable characters were parsimony-uninformative and 547 (21.61%) characters were variable and parsimony-informative. The MP analysis generated one parsimonious tree and the best tree (TL = 1,225, CI = 0.768, RI = 0.853, RC = 0.656) is presented in Figure 2. The tree topologies of ML and BI analyses were similar to that of the MP tree.

Based on the multilocus phylogeny and morphology, all three strains were assigned to one new species, named *Cytospora haidianensis*, representing a monophyletic clade with high support value (MP/ML/BI = 100/100/1).

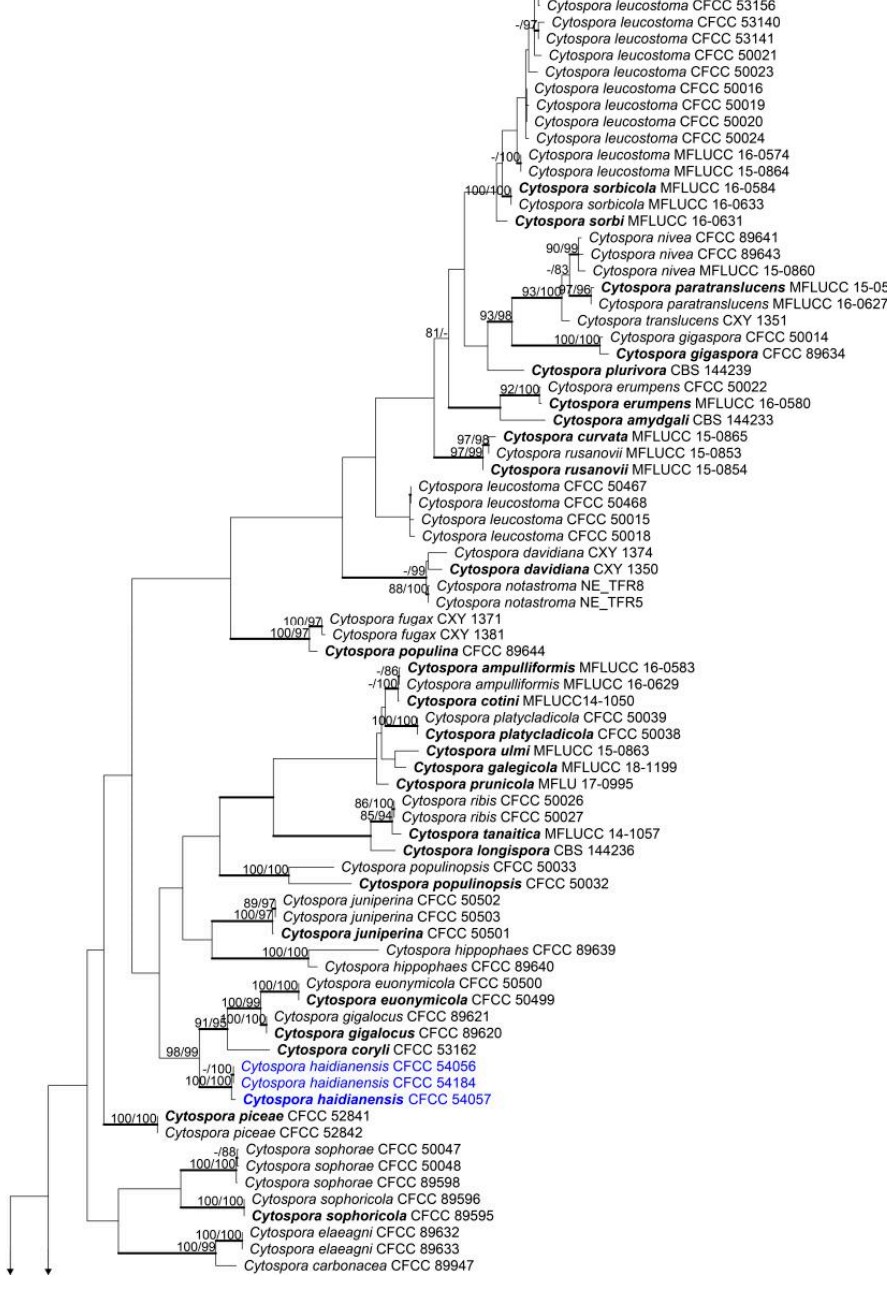

**Figure 1.** *Cont.*

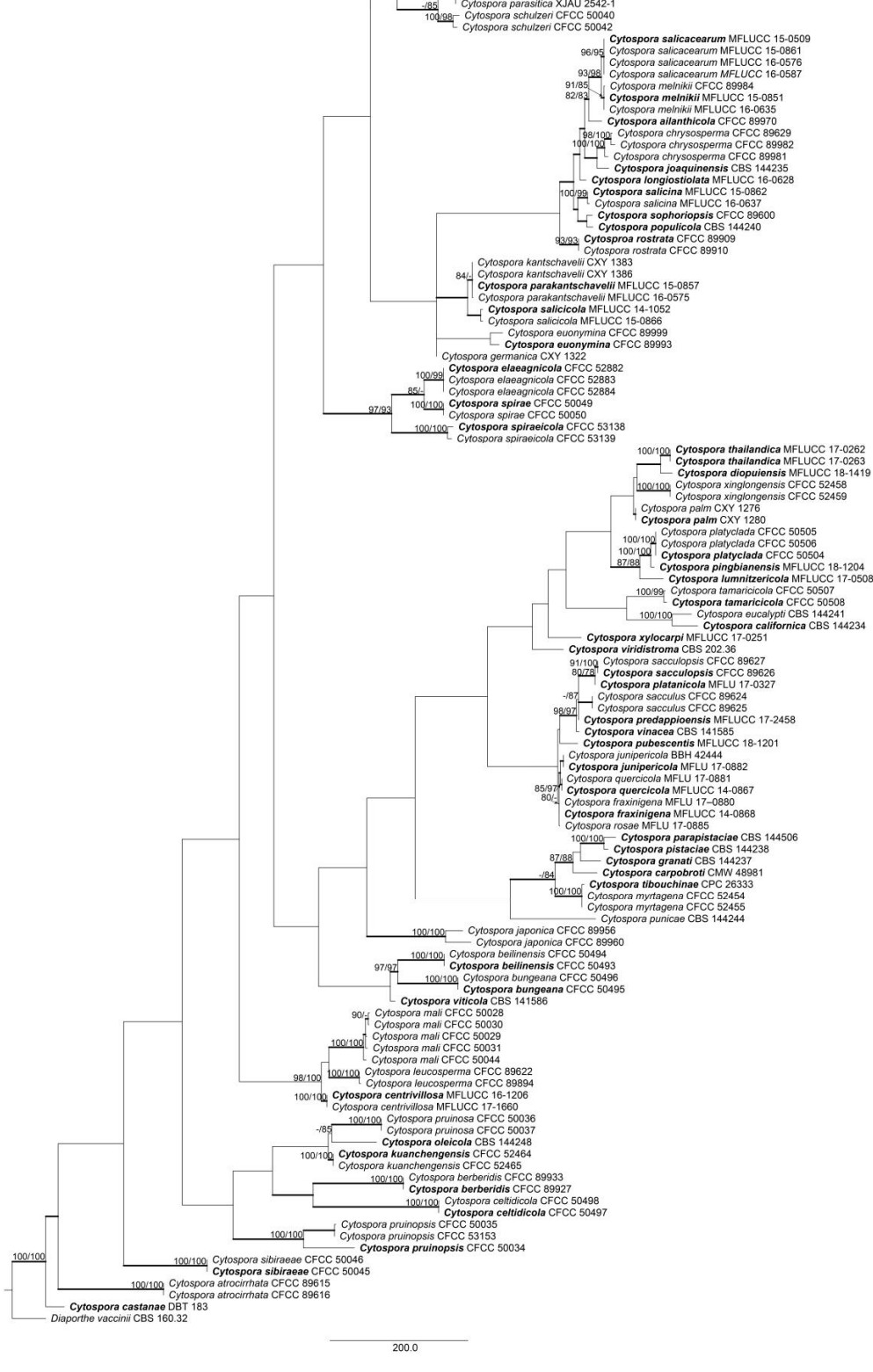

**Figure 1.** Phylogram of the best-parsimonious tree of *Cytospora* based on combined actin (ACT), internal transcribed spacer (ITS), RNA polymerase II second largest subunit (RPB2), translation elongation factor 1-alpha (TEF1-α) and beta-tubulin (TUB2) genes. Maximum parsimony (MP) and maximum likelihood (ML) bootstrap support values above 70% are shown at the first and second positions, respectively. Thickened branches represent posterior probabilities from Bayesian inference (BI) above 0.95. Scale bar = 200 nucleotide substitutions. *Diaporthe vaccinii* CBS 160.32 was used as the outgroup. Ex-type strains are in bold. Strains from the current study are in bold and blue.

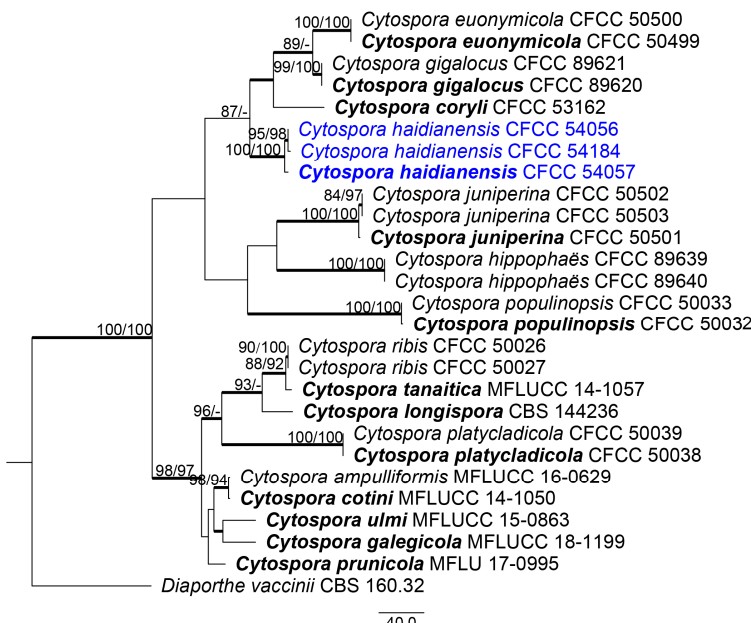

**Figure 2.** Phylogram of the best-parsimonious tree of *Cytospora* based on combined ACT, ITS, RPB2, TEF1-α and TUB2 genes. MP and ML bootstrap support values above 70% are shown at the first and second positions, respectively. Thickened branches represent posterior probabilities from BI above 0.95. Scale bar = 40 nucleotide substitutions. *Diaporthe vaccinii* CBS 160.32 was used as the outgroup. Ex-type strains are in bold. Strains from the current study are in bold and blue.

## 3.2. Taxonomy

*Cytospora haidianensis* X. Zhou & X.L. Fan, sp. nov. (Figure 3)
MycoBank MB 835121
Holotype: CF 20198643
Etymology: named after the place where it was first collected, Haidian
Host/Distribution: on cankered *Euonymus alatus* branches in China

Description: Sexual morph not observed. Pycnidial stromata ostiolate, immersed in bark, scattered, erumpent through the surface, with multiple locules. Conceptacle absent. Ectostromatic disc isabelline to dark brick, conspicuous, circular to ovoid, (330–)380–500(–520) μm ($\bar{x}$ = 460 ± 30 μm, *n* = 35) diam, with one ostiole per disc. Ostiole in the centre of the disc, black, conspicuous, (170–)179–195(–200) μm ($\bar{x}$ = 188 ± 3 μm, *n* = 10) diam. Numerous locules, subdivided frequently by invaginations with common walls, (650–)700–800(–1000) μm ($\bar{x}$ = 760 ± 30 μm, *n* = 30) diam. Conidiophores hyaline, branched at the base or unbranched, thin-walled, (9–)12–15(–16.5) × 1.0–1.5 μm ($\bar{x}$ = 13.5 ± 1.5 × 1.4 ± 0.1 μm, *n* = 50), embedded in a gelatinous layer. Conidiogenous cells enteroblastic, phialidic, subcylindrical to cylindrical, (8.5–)9–12.5(–13.5) × 1–1.5 μm ($\bar{x}$ = 11 ± 1.5 μm, *n* = 30), tapering towards the apices. Conidia hyaline, allantoid, smooth, aseptate, thin-walled, (6–)6.5–7.5 × 1–1.5 μm ($\bar{x}$ = 6.8 ± 0.2 × 1.2 ± 0.1 μm, *n* = 50).

Cultural characteristics: Colonies on PDA are initially white after 3 days, becoming light brown after 14 days. The colonies are thin with a uniform texture, lack aerial mycelium and grow up to 90 mm after 4 days. Pycnidia were randomly observed on the surface of the colony.

Material examined: CHINA, Beijing, Haidian, University Road, 116°20′19.11″ E, 40°00′16.21″ N, 51 m asl, on stems and branches of *Euonymus alatus*, Xinlei Fan, 12 November 2019 (CF 20198643, holotype; ex-type culture, CFCC 54057). Beijing, Haidian, University Road, 116°35′49.37″ E, 40°00′37.85″ N, 50 m asl, on stems and branches of *Euonymus alatus*, Xinlei Fan, 12 November 2019 (CF 20198644; living culture, CFCC 54056). Beijing, Haidian, University Road, 116°20′19.11″ E,

40°00′16.21′′ N, 51 m asl, on stems and branches of *Euonymus alatus*, Xinlei Fan, 12 November 2019 (CF 20198646; living culture, CFCC 54184).

Notes: *Cytospora haidianensis* differs from the phylogenetically related species *C. euonymicola* and *C. gigalocus* based on the sizes of the ectostromatic disc (240–350 μm diam in *C. euonymicola* and 330–620 μm diam in *C. gigalocus*), ostiole (60–120 μm diam in *C. euonymicola* and 130–190 μm diam in *C. gigalocus*), locules (1150–1400 μm diam in *C. euonymicola* and 1630–2180 μm diam in *C. gigalocus*), conidiophores (13–21.5 × 1.5–2 μm in *C. euonymicola* and 16.1–23.6 μm in *C. gigalocus*) and conidia (4.5–5 × 1 μm in *C. euonymicola* and 4.6–5.6 × 0.8–1.3 μm in *C. gigalocus*) [3,21]. Fan et al. [21] typified *C. gigalocus* based on material collected on the stems of *Juglans regia*, *C. euonymicola* and *C. euonymina* first found on twigs and branches of *Euonymus kiautschovicus* in China [3]. Similar to the other species, *C. haidianensis* also differs from the recently described species, *C. coryli*, based on macro- and micro-morphological characteristics [4]. At the molecular level, *C. haidianensis* differs from *C. euonymicola* by ACT (45/350), ITS (35/631), RPB2 (24/726), TEF1-$\alpha$ (47/725) and TUB2 (24/624), and differs from *C. gigalocus* by ACT (62/350), ITS (32/631), RPB2 (17/726), TEF1-$\alpha$ (41/725) and TUB2 (22/624).

Based on a BLAST search of the NCBI GenBank nucleotide database, the closest hits using the ACT sequence had distant hits with *Cytospora gigalocus* (strain CFCC 89620; GenBank KU710997; identities = 236/249 (94.78%), 3 gaps (1%)); *Cytospora carbonacea* (strain CFCC 50055; GenBank KP310838; identities = 237/252 (94.44%), 7 gaps (1%)). The closest hits using the ITS sequence had distant hits with *Cytospora populina* (strain CFCC 89644; GenBank KR045640; identities = 499/522 (95.59%), 10 gaps (1%)); *Cytospora cenisia* (strain CPC 28396; GenBank KY051983; identities = 489/521 (95.59%), 9 gaps (1%)). The closest hits using the RPB2 sequence had the highest similarity to *Cytospora gigalocus* (strain CFCC 89620; GenBank KU710957; identities = 690/711 (97.05%), 0 gaps (0%)); *Cytospora hippophaes* (strain CFCC 89637; GenBank KF765711; identities = 686/711 (96.48%), 0 gaps (0%)). The closest hits using the TEF1-$\alpha$ sequence had distant hits with *Cytospora coryli* (strain CFCC 53162; GenBank MN850758; identities = 397/423 (93.85%), 3 gaps (0%)); *Cytospora piceae* (strain CFCC 52842; GenBank MH820403; identities = 385/420 (91.67%), 12 gaps (2%)). The closest hits using the TUB2 sequence had distant hits with *Cytospora gigalocus* (strain CFCC 89620; GenBank KR045669; identities = 400/420 (95.24%), 11 gaps (2%)); *Cytospora leucostoma* (strain CFCC 53140; GenBank MN861115; identities = 395/419 (94.27%), 10 gaps (2%)).

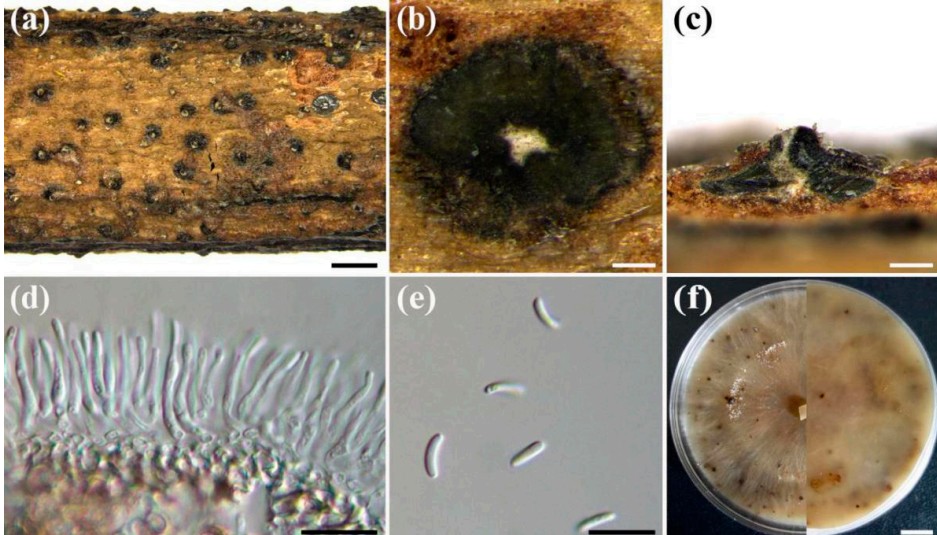

**Figure 3.** *Cytospora haidianensis* (CF 20198643). (**a**) Habitat of conidiomata on stems and branches of *Euonymus alatus*. (**b**) Transverse section of conidioma. (**c**) Longitudinal section through conidioma. (**d**) Conidiophores and conidiogenous cells. (**e**) Conidia. (**f**) Top (left) and bottom (right) sides of colonies on potato dextrose agar (PDA) after 30 days. Scale bars: **a**: 1 mm; **b**: 100 μm; **c**: 200 μm; **d**,**e**: 10 μm; **f**: 1 cm.

### 3.3. Pathogenicity Test

The three *Cytospora haidianensis* strains (CFCC 54184, CFCC 54056 and CFCC 54057) tested in this study were pathogenic on the *Euonymus alatus* twigs. No symptoms were observed in the non-inoculated controls. Brown lesions appeared at the inoculated points after 7 days of inoculation. The diseased spots turned brown and lesion areas were up to 16 mm long at 14 days after inoculation. By the third week after inoculation, the length of the brown necrotic lesions ranged from 36 to 45 mm (Figure 4). Koch's postulates were performed by successful re-isolation of fungal strains from all necrotic twigs inoculated with *Cytospora haidianensis*. The morphology and DNA sequences of the isolates retrieved from the inoculated twigs were consistent with those of the strains used for inoculation.

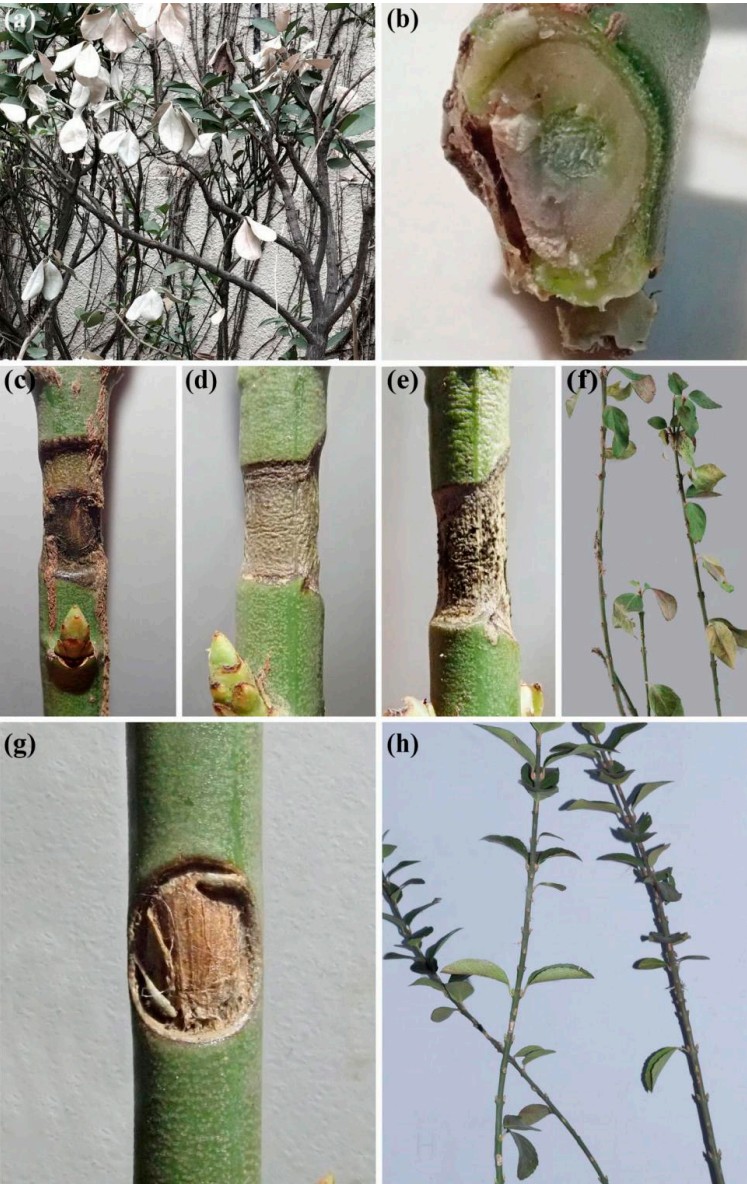

**Figure 4.** Stem blight symptoms on *Euonymus alatus* caused by *Cytospora haidianensis*. (**a**) Death of the whole plant caused by *C. haidianensis* on University Road, Beijing, China. (**b**) Stem blight caused by *C. haidianensis* in the greenhouse. Symptoms after (**c**) one week, (**d**) two weeks and (**e**) three weeks after inoculation of *C. haidianensis*. (**f**) Symptoms on *Euonymus alatus* twigs three weeks after inoculation of *Cytospora haidianensis*. (**g**,**h**) No symptoms on *Euonymus alatus* twigs after three weeks of inoculation with agar block (control).

### 3.4. Effects of Temperature and pH on Mycelial Growth

Colonies of *C. haidianensis* grew on PDA in the temperature range from 5 to 35 °C but not at 0 and 40 °C after 48 h of incubation. The fastest mycelial growth occurred at 19.8 °C, reaching 20 mm after 24 h and 86 mm after 96 h, and the least growth occurred at 5 and 35 °C. The data conform to the regression equations $Y = 4.535 + 0.986X − 0.13X^2$ ($p < 0.0001$, $R^2 = 0.846$) at 24 h, $Y = 4.747 − 2.868X − 0.64X^2$ ($p < 0.0001$, $R^2 = 0.883$) at 48 h, $Y = 6.667 + 4.821X − 0.132X^2$ ($p < 0.0001$, $R^2 = 0.868$) at 72 h and $Y = 6.263 + 8.055X − 0.239X^2 + 0.001X^3$ ($p < 0.0001$, $R^2 = 0.914$) at 96 h (X = temperature (°C), Y = growth (colony diameter, mm)). Based on the regression analysis, the optimal growth for *C. haidianensis* after incubation was estimated to occur at 19.8 °C (Figure 5).

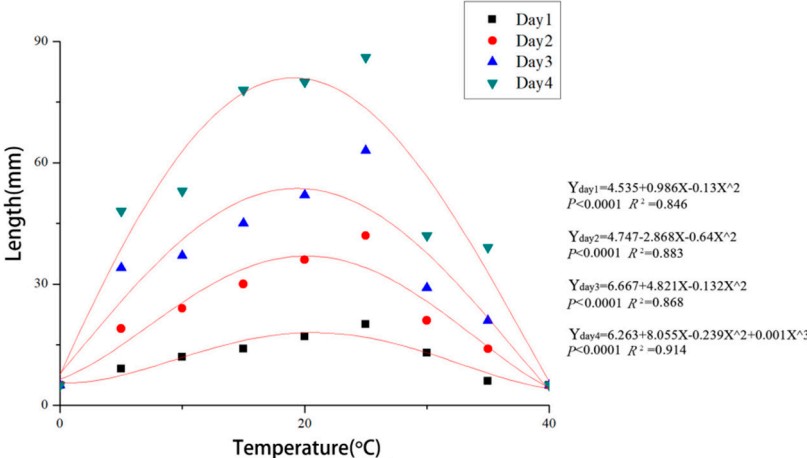

**Figure 5.** Regression curves and equations for mycelial growth of *Cytospora haidianensis* after incubation for 24, 48, 72 and 96 h at 0, 5, 10, 15, 20, 25, 30, 35 and 40 °C on PDA medium (X = temperature (°C), Y = growth (colony diameter, mm)). Optimal mycelial growth temperature was estimated to be 19.8 °C.

Colonies of *C. haidianensis* grew on PDA in the pH range from 3.0 to 10.0, but not at pH 2.0 and 12.0. After 48 h, the mycelium of *C. haidianensis* grew on PDA in the pH range from 3.0 to 10.0, but not at pH 2.0 or 12.0. Mycelium grew most rapidly at pH 9.0 after 24 h, reaching 14 mm, followed by pH 8.0 and 10.0, which gave colony diameters of 13 mm and 12 mm, respectively. The mycelia almost covered the 90 mm dishes after 96 h incubation at pH 8.0 and 9.0, while they grew more slowly at pH 3.0, 4.0, 5.0 and 11.0, attaining colony diameters of no more than 45 mm after 96 h. The data fit the regression equations $Y = 5.788 − 2.075X + 0.795X^2 − 0.53X^3$ ($p < 0.0001$, $R^2 = 0.837$) at 24 h, $Y = 10.848 − 7.209X + 2.328X^2 − 0.148X^3$ ($p < 0.0001$, $R^2 = 0.955$) at 48 h, $Y = 9.576 − 7.340X + 3.080X^2 − 0.210X^3$ ($p < 0.0001$, $R^2 = 0.964$) at 72 h and $Y = 20.424 − 17.750X + 6.382X^2 − 0.420X^3$ ($p < 0.0001$, $R^2 = 0.948$) at 96 h incubation (X = pH, Y = growth (colony diameter, mm)) (Figure 6). Based on these regression equations, the optimal growth of *C. haidianensis* after 24 and 48 h incubation was estimated to be at pH 8.3.

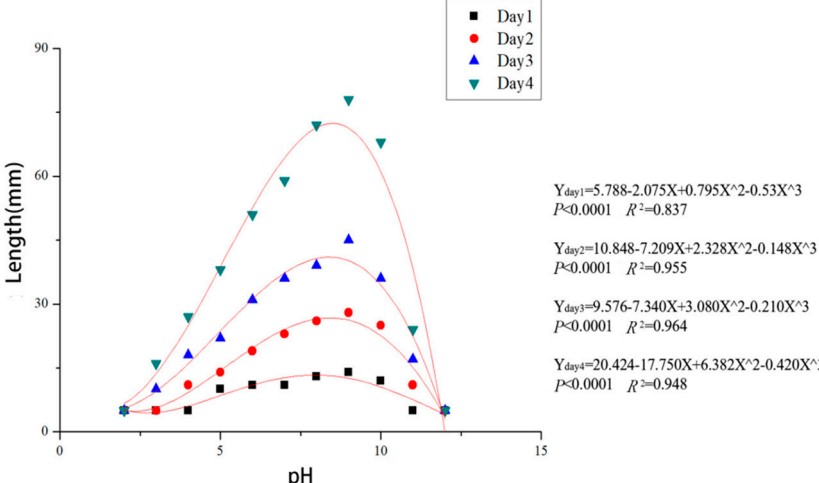

**Figure 6.** Regression curves and equations for mycelial growth of *Cytospora haidianensis* after incubation for 24, 48, 72 and 96 h at pH 2.0, 3.0, 4.0, 5.0, 6.0, 7.0, 8.0, 9.0, 10.0, 11.0 and 12.0 on PDA medium (X = pH, Y = growth (colony diameter, mm)). Optimum mycelial growth was estimated to be at pH 8.3.

### 3.5. Effects of Carbon Sources on Mycelial Growth

*Cytospora haidianensis* was able to grow using all six carbon sources tested. After 24 h, the utilization of sucrose was significantly greater than galactose, while there was no difference among fructose, glucose, xylose and maltose, which were slightly less well utilized than the other three carbon sources. The utilization of galactose was significantly lower than that of all other carbon sources tested. However, after 96 h, sucrose utilization was significantly higher than galactose and xylose, while there was no difference between fructose and glucose. Galactose had the lowest level of carbon utilization (Figure 7).

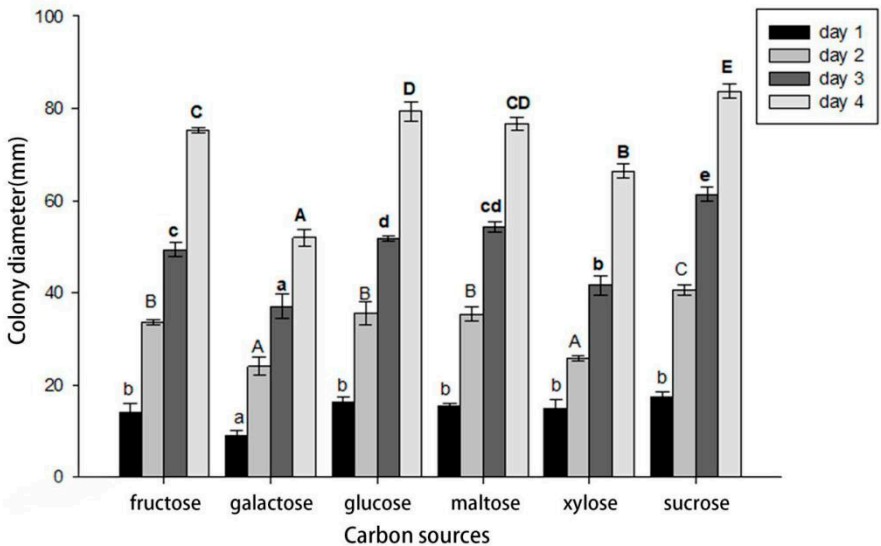

**Figure 7.** Effects of carbon source on growth of *Cytospora haidianensis*. Bars with uppercase or lowercase letters represent significant differences after, respectively, 24, 48, 72 and 96 h incubation, according to HSD tests at the $p < 0.05$ level.

## 4. Discussion

In the present study, three specimens were collected from symptomatic branches and twigs associated with dieback disease of *Euonymus alatus* in Beijing, China. A novel fungal species,

*C. haidianensis*, was introduced based on molecular, morphological and physiological data, and confirmed as the causal agent after pathogenic tests.

According to our multilocus phylogenetic analysis, *C. haidianensis* is a lineage well supported (MP-BS = 100, ML-BS = 100 and BPP = 1.0) and placed in a clade related to *C. euonymicola*, *C. gigalocus* and *C. coryli* (Figure 3). In a recent study, Fan et al. [3] described *C. euonymicola* and *C. euonymina* from twigs and branches of *Euonymus kiautschovicus* in Shaanxi Province, China. Comparing these species with the novel species *C. haidianensis*, *C. euonymicola* mainly has small ectostromatic discs (240–350 μm diam) and conidia (4.5–5 × 1 μm) and *C. euonymina* mainly has small ectostromatic discs (200–230 μm diam) and slightly larger conidia (6.5–7.5 × 1.5–2 μm), but the latter is not phylogenetically related to the new species. *Cytospora gigalocus* was described by Fan et al. [22] on stems of *Juglans regia* in Qinghai Province, China, mainly having slightly large ectostromatic discs (330–620 μm diam) and small conidia (4.6–5.6 × 0.8–1.3 μm), differing from *C. haidianensis* based on these morphological features (see notes for *C. haidianensis*). *Cytospora coryli* was recently proposed by Zhu et al. [4] as necrotrophic on branches of *Corylus mandshurica* in Mount Dongling (China), differing from *C. haidianensis* based on the size of ectostromatic discs (270–340 μm diam), large locules (1550–1710 μm diam), conidiophores (15.5–18.5 × 1–2 μm), conidiogenous cells (7.5–14 × 1–2 μm) and conidia (5–7 × 1–2 μm), and culture characteristics.

Pathogenicity tests were conducted on 1-year potted *E. alatus* plants in a greenhouse. The results indicated that *C. haidianensis* was pathogenic on *E. alatus* twigs. According to Pan et al. [7], *Cytospora* species invade the xylem and cause mortality of the whole branch, similar to the results obtained in this study within three weeks, showing the typical stem blight that occurred in the sampled place (Figure 4). The growth temperature for phytopathogenic fungi is generally from 10 to 35 °C, optimally from 20 to 30 °C [33]. For instance, the optimal growth temperature of *Penicillium cellarum* causing rot in stored sugar beet roots was reported as 22 °C [34]; for *Diaporthe neotheicola* and *D. ambigua* causing dieback blueberry in Chile, it was 25 °C; for *Diaporthe* sp., it was 22 °C [35]; and for *Phoma sorghina*, which was found to cause twisted leaf disease in sugarcane in China, it was 20–25 °C [36]. The mycelia of *C. haidianensis* grew from 5 to 35 °C, with an optimal temperature of 19.8 °C (Figure 5).

Most phytopathogenic fungi grow optimally in a pH range between 5 and 6.5 [37]. For *Lasiodiplodia vaccinii*, the range was 5.0 to 7.0, though it could still grow slowly at pH of 4.0 or 10.0 [33]. Similar results have been reported for *L. theobromae*, which could grow on media with a pH range from 4.0 to 10.0, with the optimal pH in the range of 5.0 to 7.0 [36]. The optimal pH value for *C. haidianensis* was from 8.0 to 10.0, though it could still grow slowly at pH of 4.0 or 11.0 (Figure 6). All six carbon sources tested in this study contributed to the growth of *C. haidianensis*, with less utilization of xylose than all the other carbon sources used (Figure 7).

The dieback in *Euonymus alatus* caused by *C. haidianensis* damages the plants. *Cytospora haidianensis* blights many branches and leaves discolouration, causing gradual death of a large number of *E. alatus* (Figure 4). This phenomenon is not confined to Beijing; *Cytospora euonymicola* was also reported as a pathogenic fungus from *Euonymus* in Shaanxi Province, and *Cytospora euonymina* was also found in *Euonymus* in Shanxi Province [3]. A similar phenomenon also happens in other countries; *Cytospora euonymi* was also associated with the blight of *Euonymus* twigs in the USA and Europe. Other genera such as *Cercospora*, *Colletotrichum*, *Coniothyrium* and *Fusarium* were also reported to be pathogenic fungi in *Euonymus* [38].

To date, *C. haidianensis* has been found only from *Euonymus alatus* in Beijing. Management practices, including better ventilation and lighting, might help to alleviate the damage resulting from stem dieback caused by *C. haidianensis*. The distribution and host spectrum of *C. haidianensis* need further study.

## 5. Conclusions

A novel fungal species, *Cytospora haidianensis*, is an emerging pathogen on *Euonymus alatus* dieback disease in Beijing, China. The new species is the causal agent for *E. alatus* by Koch's postulates that

grows best at 19.8 °C, pH 8.3. All the six carbon sources tested support the growth of *C. haidianensis* with the sucrose utilization is significantly higher than others.

**Author Contributions:** Experiments: X.Z., M.P. and H.L.; writing—original draft preparation: X.Z.; writing—review and editing: X.F. and C.T. All authors have read and agreed to the published version of the manuscript.

**Funding:** This study was financed by the Fundamental Research Funds for the Central Universities (2019ZY23) and College Student Research and Career-creation Program of Beijing (X201910022006).

**Acknowledgments:** We are grateful for the assistance of Xinao Mei (Beijing University of Chemical Technology), Lin Zhao (Beijing Forestry University) and Zhuang An (Shanghai Jiaotong University) during this study.

**Conflicts of Interest:** The authors declare no conflict of interest.

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
