# Peer review of "Dieback of Euonymus alatus (Celastraceae) Caused by Cytospora haidianensis sp. nov. in China"

_forests, doi:10.3390/f11050524_

Round 1
Reviewer 1 Report
The authors explain that Euonymus alatus is a plant that is widely cultivated in China as ornamental landscaping due to its capacity to adapt to stressful environmental conditions. As they mention, there is a pathogen attacking this specie and it is noticeable in many landscape areas (they mention University campuses). They also mention that Cytospora seems to be the main cause of any disease in Euonymus alatus and are interested in finding what species may be the cause of the current disease pattern they are identifying.
They outline multiple aims like I) identifying the pathogenic fungi, II) determining its pathogenicity, III) characterizing their growth at different temperatures, pH, and carbon sources.
They were able to identify a novel species of Cytospora and determine its morphology and physiological parameters.
From the data reported and their discussion, the finding of a novel specie is one of the biggest strengths of the paper. They did a good job with the methods and the step by step identification, description and overall conclusions of the characterized pathogen.
I do have some questions about details like:
-The title, which I think it is not the proper way to display their research nor highlight the finding of a novel species of pathogen here described.
-The introduction, although informative, lacks a path to go from a wide to a more narrow point of why it is important the isolation, sequencing, and characterization of the pathogen.
- I think that one main objective would make more sense instead of multiple because the separation of them is the normal step-by-step for the identification of the fungi. If any of the characteristics considered here as objectives had specific importance (maybe the carbon sources) for the country, the plant of interest or just behavior of the fungi, I think it would be important to explain and highlight it in the introduction and discussion.
-The discussion and conclusions could add some more details a highlight how some of the findings are important to the specific characteristics of the environment where the plant is and how that could impact the same plant in other locations (not just in China).
Reviewer 2 Report
Zhou et al. report isolation and characterization of a new Cytospora species that causes dieback of Euonymus alatus in China. Based on multi-locus phylogenetic analysis, the authors named the new Cytospora species as Cytospora haidianensis. Overall, the paper is scientifically sound. I only have a few minor comments shown below:
1. With respect to the virulence test, did the three isolates/strains show the same pathogenesis? Figure 3 is not cited in the Result section. Why did the authors choose CFCC 54057 as a type strain?
2. Figure 2f, what is the right half of the petri dish?
3. Typos or language issues:
L53, on the campuses
L195, typo, should be University
L241, optimal, not optimum
Reviewer 3 Report
Please find below the review of the manuscript (ms) entitled "Dieback of Euonymus alatus (Celastraceae) caused by 2 Cytospora haidianensis sp. nov. in China" by Zhou et al. submitted to forests.
In this ms, the authors identify and examine fungal isolates collected from Euonymus alatus in China and describe a novel Cytospora species. Besides multilocus phylogeny and morphological characterization of isolates of the new species, examinations of growing characteristics (pH, temperature, various carbon source utilization) were also carried out. Using Koch postulate was necessary and important to connect the microbe to the disease and it raised the value of the ms.
The ms is well written, the methods and statistics are appropriate, and the conclusions of the paper are in accordance with the results. The taxonomy part seems OK, I have couple of questions about the phylogenetic analyses). The figures should be improved in the final version! The table is not cited in the text (the order of loci is misleading here, because in the text ACT mentioned first)
The comments:
P1L2: genus and species names should be italic even in the title
P1L15: the whole names of the loci should be in the abstract
P2L71: Rayner 1970 is not cited in the list of references
P2L79: write the whole names of the loci at first occurrence, and in case of the partial coding regions the proper naming is e.g. "partial transcription-elongation factor 1-α (TEF1-α)"
P2L84: TUB should be TUB2
P3L109: 1,000,000 generation BI analysis is too weak in case of such huge data matrix (see below)
P4L156: “Cytospora species” is better
P4L166: I miss the results of the grouping of the species. It can be found very shortly in the beginning of the discussion, however basic information, especially on the “96/99 clade” comprising the novel isolates and 5 others is important
The tree (Fig 1):
I have several problems/questions about the tree. (and the quality should be better for sure)
1, why did the authors choose the MP tree instead of ML ér BI for the placement of the novel clade within the diverse Cytospora genus?
2, The absence of ML/MP bootstraps and PP values at the backbone of the tree means that using even 5 loci is not enough for some reasons to have more robust dichotomies and supported branches not counting the terminal ones. Something is not proper in the datamatrix. Therefore, this tree is quite useless, because the only information is that C. haidianensis grouped with 3 other species. Check whether everything was ok with the alignment or in the further steps!
3, I suggest that the dataset, which was used for BI should be run again using at least 10,000,000 or more generations with sampling every 1000th resulting ~10,000 trees. Why did the authors set the burn in at 25%? Increasing it to e.g. 40% may cause better congruencies. Have the topological convergences been checked by any software?
4, I suggest to the authors to incorporate a smaller tree (eg. 20-30 strains, also 5 loci) beside the big one (Fig 1) in the ms, to show the closest relatives of the three novel isolates in higher resolution, because it is much more informative than the Fig 1.
P5L168: write here what kind of tree is that
P5L169: show the BS values ≥70 on the tree!
P5L171: Only the type strain is bold
P5L171: I miss the outgroups and the description of the scale
P7L216: space before cm,
P13L320: The chapter 5 ‘conclusions’ is totally unnecessary
Reviewer 4 Report
Dear Authors,
thanks for having submitted the manuscript Dieback of Euonymus alatus (Celastraceae) caused by Cytospora haidianensis sp. nov. in China.
I have some concerns about the proposed work because there are some missing information that let me think what crtiteria you have adopted to state that Cytospora haidianensis is a new species.
In the manuscript I did not see detailed description at molecular level about the discrepancies between the isolated fungus with the closer species. Also, I did not see in the manuscript any photos of the observed symptoms but, anyway, at the beginning it was indicated that 8 universities have symptomatic Euonymus alatus but in the materials and methods is not clarified how many samples have been used for fungal isolation.
It is indicated that Cytospora haidianensis is a new species because in the phylogenic tree the merged sequences group together. This is a common result also with many strains for several plant pathogenic fungi.
In my personal opinion, if the aim of the manuscript is to describe a divergent strain of Cytospora affecting Euonymus alatus, the work has been properly conducted, described. In case the goal is to describe a new species (what I believe is the aim of the present manuscript) the manuscript can be considered after changes with a more informative description at molecular level about the differences between what the called Cytospora haidianensis and the phylogenic related species.
My kindest regards
Round 2
Reviewer 4 Report
Dear Authors,
thanks for resubmitting the paper.
I cannot find in the text any description of the sequences you obtained. To describe a new species a BLAST comparison among the available sequences and the those you got during in your work is necessary.
My Kindest regards
